# Small-Molecule-Mediated Suppression of BMP Signaling by Selective Inhibition of BMP1-Dependent Chordin Cleavage

**DOI:** 10.3390/ijms24054313

**Published:** 2023-02-21

**Authors:** Takamasa Mizoguchi, Shohei Mikami, Mari Yatou, Yui Kondo, Shuhei Omaru, Shuhei Kuwabara, Wataru Okura, Syouta Noda, Takeshi Tenno, Hidekazu Hiroaki, Motoyuki Itoh

**Affiliations:** 1Graduate School of Pharmaceutical Science, Chiba University, 1-8-1 Inohana, Chuo-ku, Chiba 260-8675, Japan; 2Graduate School of Pharmaceutical Sciences, Nagoya University, Furocho, Chikusa, Nagoya 464-8601, Aichi, Japan; 3BeCerllBar, LLC., Business Incubation Building, Nagoya University, Furocho, Chikusa-ku, Nagoya 464-8601, Aichi, Japan; 4Department of Biological Sciences, Faculty of Science, Nagoya University, Furocho, Chikusa, Nagoya 464-8602, Aichi, Japan; 5Research Institute of Disaster Medicine, Chiba University, 1-8-1 Inohana, Chuo-ku, Chiba 260-8675, Japan

**Keywords:** BMP1, BMP signaling, chordin, phenotype screening, small molecule, zebrafish

## Abstract

BMP signaling is critical for many biological processes. Therefore, small molecules that modulate BMP signaling are useful for elucidating the function of BMP signaling and treating BMP signaling-related diseases. Here, we performed a phenotypic screening in zebrafish to examine the in vivo effects of N-substituted-2-amino-benzoic acid analogs NPL1010 and NPL3008 and found that they affect BMP signaling-dependent dorsal–ventral (D–V) patterning and bone formation in zebrafish embryos. Furthermore, NPL1010 and NPL3008 suppressed BMP signaling upstream of BMP receptors. BMP1 cleaves Chordin, an antagonist of BMP, and negatively regulates BMP signaling. Docking simulations demonstrated that NPL1010 and NPL3008 bind BMP1. We found that NPL1010 and NPL3008 partially rescued the disruptions in the D–V phenotype caused by *bmp1* overexpression and selectively inhibited BMP1-dependent Chordin cleavage. Therefore, NPL1010 and NPL3008 are potentially valuable inhibitors of BMP signaling that act through selective inhibition of Chordin cleavage.

## 1. Introduction

BMP signaling is related to numerous biological phenomena. During development, the activity gradient of BMP signals is essential for dorsal–ventral (D–V) patterning and is tightly regulated by a multistep functional mechanism controlling the levels of ligands and antagonists in vertebrates [1]. BMP signaling also plays an essential role in bone formation and homeostasis [2,3].

Recently, the regulation of BMP signaling in the hippocampus was reported to be involved in the depression phenotype and aging-dependent decline in cognitive function [4,5,6]. Therefore, molecules that control BMP signaling are important for elucidating the regulatory mechanisms of BMP signaling and treating diseases caused by abnormal BMP signaling [7,8,9].

BMPs are secreted glycoproteins that belong to the transforming growth factor β (TGF-β) family. BMPs interact with two types of receptor serine/threonine protein kinases known as type I (BMPR1A, BMPR1B, and ACVR1) and type II (BMPR2, ActR2A, and ActR2B) receptors [7,8,9]. BMPs bind to the type II receptor, followed by phosphorylation of the type I receptor by the type II receptor. The phosphorylated type I receptor subsequently phosphorylates the receptor-regulated Smads (R-Smads), Smad1, Smad5, or Smad8. Phosphorylated R-Smads form a heterocomplex with a common Smad (co-Smad), Smad4, and translocate into the nucleus, where the Smad complex regulates the expression of downstream target genes of BMP signaling [7,8,9].

BMP signaling is negatively or positively modulated in several steps in its signal cascade. Among them, BMP ligand antagonists Chordin (Chrd), Noggin, Gremlin, etc., directly interact with BMP ligands and interfere with their binding to BMP receptors to negatively regulate BMP signaling [10]. Moreover, the function of Chrd is controlled by its cleavage, which is catalyzed by BMP1, a zinc-dependent metalloproteinase [11,12].

Several small molecules have been developed to artificially modulate BMP signaling and activity. Dorsomorphin, an inhibitor of BMP type I receptors, blocks the phosphorylation of Smads and inhibits BMP signaling [13,14]. UK383,367 is an inhibitor of BMP1 that suppresses BMP signaling by stabilizing the Chrd protein. However, dorsomorphin also inhibits the AMPK and Akt/mTOR pathways [15,16]. UK383,367 and their derivatives inhibit BMP1 procollagen C-proteinase activity required for collagen fibrogenesis and angiogenesis [12,17]. Thus, differences in the selectivity of inhibitors for the function of the BMP signaling pathway might lead to broader applications.

In previous research, we screened Disheveled1(Dvl1)-PDZ domain inhibitors by performing virtual screening and identified N-substituted-2-amino-benzoic acid analogs [18]. We further explored the effect of N-substituted-2-amino-benzoic acid analogs in vivo by performing a phenotype screening in zebrafish. In the present study, we found that NPL1010 and NPL3008 are potential BMP signaling inhibitors that selectively suppress BMP1-mediated Chrd cleavage activity.

## 2. Results

### 2.1. NPL1010 and NPL3008 Induced a Distinct Phenotype

We performed phenotype screening using zebrafish embryos to explore N-substituted-2-amino-benzoic acid analogs that had been identified from in silico screening for their capacity to bind to the PDZ domain of Dvl1. Among them, ten candidate compounds (Appendix A) were applied to zebrafish embryos at 3 h post-fertilization (hpf) using several concentrations for 8 or 21 h, and their effects on development were then examined.

NPL1010 and NPL3008 induced animal–vegetal (A-V) axis elongated morphology at 11 hpf (Figure 1A,B: 25 μM NPL1010, 81% (*n* = 36) with an elongated A-V axis phenotype; 100 μM NPL1010, 82% (*n* = 34) with an elongated A-V axis phenotype; 2.5 μM NPL3008, 10% (*n* = 10) with an elongated A-V axis phenotype; and 5 μM NPL3008, 70% (*n* = 10) with an elongated A-V axis phenotype). At 24 hpf, NPL1010 or NPL3008 treated embryos exhibited reductions in the ventral tail fin and tail length at 24 hpf. These phenotypes are characteristic features of dorsalization [19]. We classified the dorsalized phenotypes into previously reported C1−C5 categories [19] (Appendix A). NPL1010 induced a severe phenotype at both 25 and 100 μM treatment. The severity of the phenotype induced by NPL3008 was dose-dependent (Figure 1C,D: 25 μM NPL1010, 86% (*n* = 28) with a dorsalized C1−C5 phenotype; 100 μM NPL1010, 76% (*n* = 25) with dorsalized C1−C5 phenotype; 2.5 μM NPL3008, 60% (*n* = 10) with dorsalized C1−C5 phenotype; and 5 μM NPL3008, 90% (*n* = 10) with dorsalized C1−C5 phenotype).

Other compounds resulted in high lethality and/or did not induce specific morphological phenotypes (Figure 1B,D). NPL 1010 and NPL3008 share a common structure, 2-(3-benzamidobenzamido) benzoic acid, but the length and position of the ether groups differ: NPL1010 contains an ethoxy group in the para, whereas NPL3008 contains a propoxy group in the meta position (Figure 1E). On the other hand, NPL3005 and NPL3006 also contain 2-(3-benzamidobenzamido) benzoic acid, similar to NPL1010 and NPL3008, but they did not induce the elongated A-V axis phenotype or dorsalized phenotype. Based on these results, the 2-(3-benzamidobenzamido) benzoic acid structure is necessary but not sufficient to induce the elongated A-V axis phenotype and dorsalized phenotype; in addition, the ether group on benzene might play a role in inducing the morphological changes.

Next, we further explored the effects of their analogs, which are commercially available, on zebrafish embryo morphogenesis. First, we modified the ether group; a change in the length of the carbon chain attached to the ethoxy group of NPL1010 to a propoxy group (NS00479226) did not enhance the effect of NPL1010. In addition, the change from an ether to an acetate group (NS02237411) did not improve the effect of NPL1010 (Appendix A).

Second, we modified the benzoic acid groups of NPL1010 and NPL3008. However, any tested modification of the benzoic acid group of NPL1010 and NPL3008 did not enhance their effects on inducing the elongated A-V axis phenotype shape and dorsalized phenotype (Appendix A).

Based on these data, the benzoic acid group and the position and length of the ether group are critical for the mechanisms of action of NPL1010 and NPL3008.

### 2.2. NPL1010 and NPL3008 Induce a Dorsalized Phenotype

NPL1010 and NPL3008 induced phenotypes of elongated A-V, reduction in the ventral tail fin, and short tail length, which are characteristic phenotypes of dorsalization [19,20]. To confirm the expansion of dorsal structure by changes in gene expression patterns, as previously reported [21,22], we performed double in situ hybridization with *myoD* and *krox20* probes, which stain the presomitic mesoderm (Appendix A, black arrowheads) and rhombomeres 3 and 5, respectively (Appendix A, magenta arrowheads). Compared with DMSO-treated embryos, NPL1010- and NPL3008-treated embryos exhibited lateral expansion of the *krox20*-positive domain and *myoD* expression domain, indicating that the embryos have an enlarged dorsal structure at 11 hpf (Appendix A). These results indicate that NPL1010 and NPL3008 induce the dorsalization of zebrafish embryos.

### 2.3. NPL1010 and NPL3008 Reduce the Expression of Ventral Genes

During vertebrate development, the proper regulation of dorsal and ventral gene expression is required [23]. Body axis dorsalization is induced by an increase in the expression of dorsal genes or a decrease in the expression of ventral genes [24]. We explored these possibilities by performing in situ hybridization of the dorsal-specific gene *chrd* and the ventral gene *ved* (Figure 2A–D). In dorsalized embryos, the dorsal expression domain of *chrd* was expanded [21,25,26]. Therefore, we measured the central angle of the *chrd* expression domain as previously reported [25]. *chrd* expression domain was significantly expanded by NPL1010 and NPL3008 treatments (Figure 2A,B and Appendix A). In contrast, NPL1010- or NPL3008-treated embryos exhibited a reduction in *ved* expression (+ or ++ in Figure 2C,D) compared with that in DMSO-treated controls (normal in Figure 2C,D). In addition, expression of another ventral marker gene *szl* [27,28] was also reduced by NPL1010- or NPL3008 treatment (Appendix A). Furthermore, the decrease in ventral *ved* and *szl* expression induced by these two compounds was dose-dependent (Figure 2D and Appendix A). Previous reports indicated that *ved* and *szl* expression domain was reduced in dorsalized embryos [29,30]. Therefore, NPL1010 and NPL3008 induce dorsalization by inhibiting ventral gene function.

### 2.4. NPL1010 and NPL3008 Inhibit Bone Formation In Vivo

NPL1010 and NPL3008 were originally obtained as Dvl1-PDZ domain inhibitors through virtual screening [18]. Therefore, we examined the effects of NPL1010 and NPL3008 on Wnt signaling. NPL1010 and NPL3008 did not affect TOP flash activity in HEK293T cells (Appendix A). These data indicate that NPL1010 and NPL3008 may not necessarily inhibit Wnt/βcatenin signaling.

A reduction in ventral gene expression is known to be associated with reduced BMP signaling, as observed in BMP signaling mutants such as *swirl* (*bmp2b* mutant) [31]. Because BMP signaling is important for bone mineralization, the inhibition of BMP signaling affects the mineralization of bone [32]. We performed Alcian blue and Alizarin red double staining to assess the effects of NPL1010 and NPL3008 treatments on bone formation in vivo (Figure 3A).

We investigated the effects of the two compounds on parasphenoid bone mineralization, in which shape and mineralization are readily observed (Figure 3B, white dotted lines). We classified bone mineralization into three categories: normal mineralization, slight reduction in mineralization/mild phenotype (+), and strong reduction in mineralization/severe phenotype (++) (Figure 3B).

Treatment with the BMP inhibitor dorsomorphin, an inhibitor of BMP type I receptors, reduced bone mineralization (Figure 3C), consistent with previous reports [13,32]. Similarly, NPL1010 and NPL 3008 treatment decreased the percentage of embryos with mineralized bone (Figure 3C). These data suggest that NPL1010 and NPL 3008 are inhibitors of BMP signaling that modulate BMP signaling and activity in vivo.

### 2.5. NPL1010 and NPL3008 Decrease Activity of the BMP Signaling Reporter

Next, we further explored the possibility that NPL1010 and NPL3008 inhibit BMP signaling by first performing a culture-cell-based BMP reporter dual-luciferase assay [33,34]. Overexpression of *bmp2b*, a BMP ligand, substantially increased the activity of the BMP signaling reporter in C2C12 cells (Figure 4A). The known BMP signaling inhibitor dorsomorphin suppressed this effect of *bmp2b* (Figure 4A). NPL1010 and NPL3008 also reduced *bmp2b*-induced BMP signaling activity in a dose-dependent manner (Figure 4A). We next examined the effects of the two compounds downstream of the BMP receptor. Alk6 is a BMP signaling receptor. Smad1 and Smad5 are BMP signaling mediators that function downstream of BMP receptors. Smad1 and Smad5 are phosphorylated by activated BMP receptors, form a complex with Smad4, translocate to the nucleus, and regulate downstream gene expression [7,8,9].

Overexpression of the constitutively active form of Alk6 (CA-Alk6), Smad1, and Smad5 activated BMP signaling, as previously reported [35] (Figure 4B). Dorsomorphin treatment significantly suppressed Smad1, Smad5, and CA-Alk6-dependent activation of BMP signaling (Figure 4B). On the other hand, NPL1010 and NPL3008 rarely inhibited BMP signaling activated by CA-ALK6 or the combined expression of Smad1 and Smad5 (Figure 4B). These results indicate that NPL1010 and NPL3008 act at a step upstream of the BMP receptors.

### 2.6. NPL1010 and NPL3008 Suppress BMP1a-Dependent Chrd Cleavage

We performed rescue experiments to further clarify the mechanisms of action of NPL1010 and NPL3008 in the BMP signaling pathway. mRNAs for *bmp2b* or genes encoding the BMP signaling modulators, *bmp1a* and *tll1*, which are metalloproteases that positively regulate BMP signaling by cleaving the BMP antagonist Chrd [11], were injected, and they induced the ventralized phenotype (Figure 5A–D), consistent with previous reports [11]. Dorsomorphin inhibited *bmp2b-*, *bmp1a-*, and *tll1*-dependent ventralization, as expected [13]. On the other hand, NPL1010 and NPL3008 partially suppressed *bmp1a*-induced ventralization (Figure 5A–D). In addition, NPL1010 and NPL3008 also weakly suppressed *tll1*-induced ventralization, although less efficiently than suppression of *bmp1a*-iuduced ventralization. NPL1010 and NPL3008 did not strongly inhibit *bmp2b*-induced ventralization in zebrafish embryos, unlike in C2C12 cells. This discrepancy might be due to the different ratios of exogenous *bmp2b*/endogenous *bmp1a* in embryos than in C2C12 cells, which express high levels of Chrd and BMP1 [25,36]. These results suggest that NPL1010 and NPL3008 are inhibitors of BMP1 that degrade the BMP antagonist Chrd [12,37,38]. In zebrafish, *bmp1a* mutants or morphants exhibit ruffled fins [39,40]. This might be due to the reduction in collagen maturation caused by Bmp1a-dependent procollagen cleavage. However, embryos treated with NPL1010 or NPL3008 did not show the ruffled phenotype (Appendix A). These data suggest NPL1010 and NPL3008 might selectively inhibit BMP1-dependent Chrd cleavage. Therefore, we next examined the effect of these compounds on BMP1-mediated cleavage of Chrd.

We transfected the C-term Myc-tagged Chrd expression vector into HEK 293T cells and prepared Chrd-Myc-containing conditioned medium. We confirmed that Myc-tagged Chrd was functional in zebrafish embryos because it induced the dorsalization phenotype (Appendix A). The addition of recombinant human BMP1 (rhBMP1) to Chrd-Myc-containing conditioned medium accelerated the cleavage of Chrd and increased the ratio of Chrd cleaved at the C-terminus to total Chrd (Figure 5E,F). NPL1010 or NPL 3008 treatment increased the abundance of full-length and intermediate part + C-term fragments of Chrd (Figure 5E). We found that NPL1010 or NPL3008 inhibit rhBMP1-dependent cleavage of Chrd in a dose-dependent manner (Figure 5E,F).

Chrd is one of numerous substrates of BMP1, and we assessed whether NPL1010 and NPL3008 inhibit the cleavage of other BMP1 substrates. Collagen is also cleaved by BMP1. Although the fin phenotype suggests that NPL1010 and NPL3008 might not affect collagen cleavage (Appendix A), we examined the effects of NPL1010 and NPL3008 on BMP1-dependent collagen type III alpha1 (COL3A1) cleavage. UK383,367, a BMP1 inhibitor, strongly interfered with BMP1-dependent COL3A1 cleavage (Figure 6A,B). On the other hand, NPL1010 and NPL3008 rarely inhibited BMP1-dependent COL3A1 cleavage (Figure 6A,B).

### 2.7. NPL1010 and NPL3008 Potentially Bind to the Cleft in the BMP1 Catalytic Domain

Since the NPL compounds used in this study were initially identified as common molecular scaffolds for PDZ domain inhibitors based on their chemical structures, their potential to bind BMP1 was assessed by performing docking simulation experiments. After the visual assessment of the docking poses, we found that the docking pose of NPL1010 with the lowest energy and that of NPL3008 with the second-lowest energy were quite similar, wherein both bind to the shallow bottom of the ligand binding cleft of BMP1 near the Zn^2+^ atom in the catalytic center (Appendix A). Although the configuration of ether groups differed between NPL 1010 and 3008, the dihedral angle of the carboxyl group of the ring was almost inverted in their docked poses. As a result, the positions of these ether groups were similar (Appendix A). These simulation results show that NPL1010 and NPL3008 bind at a location near Asn128/Thr156/Phe157 of the BMP1 catalytic domain (Appendix A). However, this putative interaction site for NPL1010 and NPL3008 with BMP1 is different from the interaction sites of known inhibitors Compound 1 (PDB: 6BTN), Compound 4 (PDB: 6BSM), and Compound 22 (PDB: 6BSL) reported in a previous study [41]. In addition, the simulated interaction between the BMP1 catalytic domain and the Chrd C-terminal region (241 aa) produced using AlphaFold2 [42,43] demonstrated that the Chrd C-terminal cleavage site (PMQADGPR) is closely located near Thr156/Phe157 of BMP1 and that its position overlaps with the putative NPL1010 and NPL3008 interaction sites (Appendix A). These results suggest that NPL1010 and NPL3008 inhibit BMP1 activity through different mechanisms than those of previously reported inhibitors [17,41].

## 3. Discussion

Our results show NPL1010 and NPL3008 can suppress BMP signaling activation via selectively inhibiting BMP1-dependent Chrd cleavage, unlike inhibitors of BMP type I receptors, such as dorsomorphin [13,14] (Figure 7). Several compounds have been reported as BMP1 inhibitors, such as UK383,367 and its derivatives. These compounds contain hydroxamic acid or reverse hydroxamic acid group structures. These structures inhibit the enzymatic activity of BMP1 by binding to zinc ions in the active center of BMP1 [17,41]. On the other hand, NPL1010 and NPL3008 do not contain these functional groups in their structures and therefore inhibit BMP1 function through a different mechanism from that of known BMP1 inhibitors. Both NPL1010 and NPL3008 contain the 2-(3-benzamidobenzamido) benzoic acid moiety, but the ether group length and position differ: NPL1010 contains an ethoxy group at the para position, whereas NPL3008 contains a propoxy group at the meta position. Our phenotype screening experiments indicated that modification of the 2-(3-benzamidobenzamido) benzoic acid structure or the length and position of the ether group did not enhance the effects of NPL1010 and NPL3008. Therefore, both 2-(3-benzamidobenzamido) benzoic acid and additional ether groups, namely para-ethoxy groups and meta-propoxy groups, are required for the strong inhibition of BMP1 function. The docking simulation shows that the ether group orientation is inverted between NPL1010 and NPL3008. The positions of these ether groups are similar. Thus, the three-dimensional angle and length of the ether group may be important for the interaction with BMP1, which may explain why other NPL1010 and NPL3008 derivatives that do not contain para- or meta-ether groups, such as NPL3005 and NPL3006, did not show inhibitory activity, in contrast to NPL1010 and NPL3008.

UK383,367 and its derivatives also inhibit the cleavage of BMP1 substrates other than Chrd, such as procollagen [17]. In contrast, NPL1010 and NPL3008 did not inhibit the cleavage of COL3A1, which is a collagen family member and a BMP1 substrate other than Chrd [12,44]. Thus, NPL1010 and NPL3008 show high selectivity for the inhibition of Chrd cleavage by BMP1 (Figure 7), although the effects of NPLs on BMP1 substrates other than procollagen need to be investigated in future.

NPL1010 and NPL3008 selectively inhibit BMP1-dependent Chrd cleavage and inhibit BMP signaling in a Chrd function-dependent manner. Known inhibitors (e.g., UK383,367) inhibit cleavage of both Chrd and collagen. Dorsomorphin blocks the phosphorylation of Smads and inhibits BMP signaling.

The high substrate selectivity of NPL1010 and NPL3008 over UK383,367 may suggest that NPL1010 and NPL3008 do not inhibit the enzymatic activity of BMP1 itself but instead interfere with the interaction between BMP1 and Chrd. According to previous studies, the catalytic domain and CUB1 domain are required for Chrd cleavage but are not sufficient for procollagen cleavage; the CUB2 domain of BMP1 is required for procollagen cleavage [45]. Differences in the substrate recognition mechanism might contribute to determining the selectivity of NPL1010 and NPL3008 for Chrd. In addition, a previous structural analysis suggested that the CUB1CUB2 (C1C2) fragment of procollagen C-proteinase enhancer-1 (PCPE-1) binds to the C-propeptide trimer of procollagen III and pulls the procollagen chain toward C1C2. Then, the procollagen chain may enter the active site of the BMP1 catalytic domain, where the P1′ residue Asp interacts with Arg182 in the S1′ pocket in close proximity to the essential Glu94 and the catalytic water molecule bound to the active site zinc ion [46]. However, PCPE-1 does not enhance BMP1-dependent Chrd cleavage [45]. In addition, the putative interaction site for NPL1010 and NPL3008, comprised of BMP1 Asn128/Thr156/Phe157, seems to differ from that between BMP1 and procollagen [46]. Our interaction simulation demonstrated that the Chrd C-terminal cleavage site is located in close proximity to Thr156/Phe157 of BMP1. Therefore, Thr156/Phe157 may be required for the interaction between BMP1 and Chrd but not for the interaction between BMP1 and procollagen. These differences may contribute to the selectivity of NPL1010 and NPL3008, although future studies are needed to validate this model. Our simulation results suggest that NPL1010 and NPL3008 might interact with the putative Chrd-interacting site of the BMP1 catalytic domain. Therefore, NPL1010 and NPL3008 might function as reversible and competitive inhibitors. To address this possibility, further studies are needed.

*bmp1a* mutants and morphants do not show the dorsalized phenotype [39,40]. However, *bmp1a* and *tll1* double knockdowns show the dorsalized phenotype [40]. Our data indicate NPL1010 and NPL3008 might also affect Tll1 function. The catalytic domain is highly conserved between BMP1 and Tll1 [47]. In addition, the putative NPL1010 and NPL3008 interaction site comprising Asn128/Thr156/Phe157 is also conserved between BMP1 and Tll1. These indicate that NPL1010 and NPL3008 may inhibit BMP1 and Tll1 function by the same molecular mechanisms, and dorsalized phenotypes induced by NPL1010 or NPL3008 are due to suppression of BMP signaling by both BMP1 and Tll1 inhibition.

Our data suggest that NPL1010 and NPL3008 can suppress BMP signaling activation by inhibiting BMP1-dependent Chrd cleavage. However, we cannot rule out the possibility that NPL1010 and NPL3008 might also inhibit the maturation or secretion of BMP ligands or induce a conformational change in BMP receptors. Further studies are needed to investigate the possible mechanisms by which NPL10101 and NPL3008 inhibit BMP signaling.

We demonstrated that NPL1010 and NPL3008 can prevent human BMP1-dependent Chrd cleavage. Although the efficiency and pharmacokinetics of NPL1010 and NPL3008 in human body must be investigated, NPL1010 and NPL3008 may be applied in the treatment of abnormal BMP signaling causing human diseases, such as the depression phenotype and aging-dependent decline in cognitive function, by upregulating the functional Chrd protein level [4,5,6].

In conclusion, NPL1010 and NPL3008 provide additional options to regulate BMP signaling. The combination of NPL1010 and NPL3008 with existing inhibitors of BMP signaling may widen the therapeutic window for diseases caused by abnormal BMP signaling [4,5,6,7,8,9].

## 4. Materials and Methods

### 4.1. Screening of N-Substituted-2-Amino-Benzoic Acid Analogs

N-Substituted-2-amino-benzoic acid analogs were screened using a virtual screening method based on the structure of the Dvl1-PDZ domain [18].

### 4.2. Preparation of Stock Solutions of Compounds and Protein

Supplier and catalog numbers are shown in Appendix A for the chemical compounds used in the structure–activity relationship experiment. The chemical compounds were dissolved in DMSO to 100 mM and stored at −30 °C. Dorsomorphin (LC Laboratories, Woburn, MA, USA, catalog no. D-3197) was dissolved in DMSO to 25 mM and stored at −30 °C. Recombinant human BMP1 (rhBMP1, 0.276 mg/mL, dissolved in 25 mM HEPES containing 400 mM ammonium sulfate, R&D Systems, Minneapolis, MN, USA, catalog no. 1927-ZN-010) and recombinant human COL3A1 (rhCOL3A1, 0.336 mg/mL, dissolved in 25 mM sodium acetate containing 1 M NaCl, R&D Systems, catalog no. 7294-CL-020) were divided into aliquots and stored at −80 °C.

### 4.3. Dual-Luciferase Reporter Assay

For measuring BMP signaling activity, C2C12 cells were seeded into 96-well plates at a density of 5 × 10^3^ cells/well in DMEM (10% FBS, 1% penicillin streptomycin (PS)) and cultured for 24 h. pCS2+ full length-bmp2b or pCS2+ (100 ng), 50 ng of pGL4-BRE-fluc, and 4 ng of pGL4.74 [hRluc/TK] were transfected into the cells in each well using polyethyleneimine (PEI). In the CA-Alk6 and Smad overexpression experiment, 20 ng of pCDNA3-CA-Alk6, 15 ng of pcDNA3-FLAG-Smad1, and/or 15 ng of pcDNA3-FLAG-Smad5 were also transfected with PEI. For measuring Wnt signaling activity, HEK293 cells were seeded into 96-well plates at a density of 3 × 10^4^ cells/well in DMEM (10% FBS, 1% penicillin streptomycin (PS)) and cultured for 24 h. Twenty nanograms of pCS2+ mouse Wnt1 or pCS2+, 70 ng of TOP Flash or FOP Flash reporter plasmid, and 4 ng of pGL4.74 [hRluc/TK] were transfected into the cells in each well using polyethyleneimine (PEI). After plasmid transfection, the cells were incubated at 37 °C with 5% CO_2_ for 3 h. NPL1010 and NPL3008 were added at 5, 12.5, 25, and 50 μM, and dorsomorphin was added at 1, 5, 12.5 μM. After 21–24 h of incubation at 37 °C with 5% CO_2_, the culture medium was removed, the cells were washed with 100 µL/well of 1× PBS, and 25 µL of 1× Passive Lysis Buffer (5× Passive Lysis Buffer (Promega, Madison, WI, USA) diluted to 1/5) was added to each well. The cells were lysed by shaking at room temperature for 20 min, and the lysates were transferred to 1.5 mL tubes. After centrifugation at 15,000× *g* for 2 min at 4 °C, 10 µL of the supernatant were transferred to a white 96-well plate (Thermo Fisher Scientific, Waltham, MA, USA). Fluc- and hRluc-dependent luminescence was measured by sequentially adding 40 µL/well of luciferin buffer (50 mM Tris-HCl, pH 7.8, 10 mM MgCl_2_, 500 µM coenzyme A, 300 µM ATP, and 200 µg/mL D-luciferin K) and 50 µL/well of Renlite buffer (0.04 mM PTC124, 6.7 μM coelenterazine h, 30 mM Na EDTA, 20 mM Na pyrophosphate, and 950 mM NaCl) and detected using the GloMax^®^-Multi Detection System (Promega). Each sample was measured in duplicate. The Fluc activity was normalized to the hRluc activity.

### 4.4. Preparation of Chrd-Containing Conditioned Medium

HEK293T cells were seeded on collagen (Cellmatrix type I-C, Nitta Gelatin, Yao, Japan)-coated 10 cm dishes at a density of 3.0 × 10^6^ cells/dish, simultaneously reverse-transfected with 20 µg of pCS2+ Chrd-Myc [48], and incubated at 37 °C with 5% CO_2_ for 24 h. After incubation, the medium was removed, cells were washed with 1× PBS, and the medium was replaced with Opti-MEM (Thermo Fisher Scientific). After 48 h of incubation under the same conditions, the culture supernatant was collected and stored at −80 °C.

### 4.5. Chrd Cleavage Assay

For this experiment, 18.4 µL of OptiMEM (Thermo Fisher Scientific) was added to a 1.5 mL tube, and 18 µL of conditioned medium containing Chrd-Myc was added to each tube. A stock solution of each compound was diluted with DMSO to 5, 10, 25, and 50 mM. The solution of each compound was further diluted 1:40 with Opti MEM, and 1.6 µL of the diluted solution was added to each sample. The 10 mg/mL rhBMP1 stock solution was diluted 1:10 with OptiMEM, and 2 µL of diluted rhBMP1 were added to each sample and mixed. After incubation at 37 °C for 1 h, an equal volume (40 µL) of 2× sample buffer (125 mM Tris-HCl, pH 6.8, 4% SDS, 20% glycerol, 10% 2-mercaptoethanol, and 0.001% bromophenol blue) was added to the samples, and they were then heated at 98 °C for 5 min.

### 4.6. Western Blotting

The prepared samples were separated on 10% sodium dodecyl sulfate (SDS)–polyacrylamide gel electrophoresis (PAGE) gels and transferred to PVDF membranes (Merck, Darmstadt, Hesse, Germany). The membranes were blocked with 0.3% skim milk (FUJIFILM Wako, Osaka, Japan) in Tris-buffered saline containing 0.1% Tween 20 (TBST) for 1 h and incubated with an anti-c-Myc mouse monoclonal antibody (1:2000, FUJIFILM Wako, 9E10, catalog no. 011-21874) overnight at 4 °C. Then, the membranes were washed three times with TBST and incubated with HRP-conjugated goat anti-mouse IgG (H + L) (1:20,000, Jackson Immunoresearch, West Grove, PA, USA, catalog no. 115-035-003) at RT for 1 h. The membranes were washed three times with TBST and incubated with the ECL solution (Solution 1: 2.5 mM luminol, 4 mM 4-IPBA, and 200 mM Tris-HCl, pH 8.8; Solution 2: 10.6 mM H_2_O_2_). Equal volumes of each solution were mixed immediately before use. Membranes were imaged using an EZ-capture MG instrument (ATTO, Tokyo, Japan). Quantitative densitometry was used to measure the levels of protein fragments with a CS Analyzer (ATTO).

### 4.7. COL3A1 Cleavage Assay

Recombinant human procollagen 3A1 (1.41 ng, R&D, catalog no. 7294-CL-020) was mixed with 4–8 ng of recombinant human BMP1 and leupeptin (final 20 µM, Nacalai Tesque, Kyoto, Japan, catalog no. 20454-76) in assay buffer (25 mM HEPES-NaOH, pH 7.5, and 0.01% Brij35). NPL1010, NPL3008, or UK383,367 was added to the mixture (the final concentrations of each compound were 10, 25, or 50 µM). The total volume of the reaction mixtures was adjusted to 20 µL. The mixtures were incubated at 37 °C for 1 h, an equal volume (20 µL) of 2× sample buffer was added to the mixtures, and mixed samples were heated at 98 °C for 5 min. Heated samples were separated on 10% SDS–PAGE gels and stained with Oriole Fluorescent Gel Stain (Bio–Rad, Hercules, CA, USA, catalog no. 1610496) according to the manufacturer’s instructions. Stained gels were imaged using ImageQuant LAS4000 (Cytiva, Marlborough, MA, USA), and the relative amount of proteins was analyzed using CS Analyzer (ATTO). BMP1-dependent cleavage of COL3A1 was calculated from the amount of each COL3A1 fragment by subtracting BMP1-independent degradation of COL3A1 based on the molecular weight of each fragment.

### 4.8. Fish Maintenance

Zebrafish were raised and maintained under standard conditions with the approval of the Chiba University Institutional Animal Care and Use Committee (Nos. 1-174, 2-178, 3-66, and 4-11). Zebrafish embryos were obtained from the natural spawning of wild-type adults.

### 4.9. Phenotypic Screening

One to five zebrafish embryos were placed in 24-well plates with 1 mL of E3 medium (5 mM NaCl, 0.17 mM KCl, 0.33 mM CaCl_2_, and 0.33 mM MgSO_4_) and exposed to several concentrations of compounds from 3 to 11 hpf or 3 to 24 hpf; the phenotype was observed after incubation. Images were captured using a stereomicroscope (Leica, Wetzlar, Hesse, Germany, MZ16) and Axio Cam MRc (Zeiss, Oberkochen, Baden-Württemberg, Germany). For examination of the NPL1010 and NPL3008 effects on fin morphogenesis and bone mineralization, the chorion was removed at 1 dpf, and embryos were treated with NPL1010 or NPL3008 from 1 dpf to 3 or 5 dpf.

### 4.10. RNA Probe

Each RNA probe was previously described. The cDNA fragments for *chrd* [26], *ved* [29], *szl* [27,28], *krox20* [49], and *myoD* [50] were utilized as templates for the antisense probes. The antisense probes were synthesized with T3, T7 and Sp6 RNA polymerase using linearized templates.

### 4.11. Whole-Mount In Situ Hybridization

The embryos were collected in a 1.5 mL tube and fixed with 4% paraformaldehyde (PFA)/1× phosphate-buffered saline (PBS) at 4 °C overnight. The embryos were dechorionated and dehydrated by sequential treatment with 25% methanol/1× PBS supplemented with 0.1% Tween 20 (1× PBSTw), 50% methanol/1× PBSTw, and 75% methanol/1× PBSTw. The embryos were incubated with each solution for 5 min at room temperature (RT). Then, the embryos were treated with 100% methanol at −30 °C for 30 min. The embryos were hydrated by sequential treatment with 75% methanol/1× PBSTw, 50% methanol/1× PBSTw, and 25% methanol/1× PBSTw for 5 min at RT each. The embryos were washed with 1× PBSTw for 5 min at RT twice and prehybridized with hybridization buffer (HYB, 50% formamide, 5× saline sodium citrate (SSC), 0.1% Tween 20, and 50 µg/mL heparin) at 65 °C for 1 h. Then, the prehybridization HYB was replaced with HYB containing a probe, and the embryos were hybridized at 65 °C overnight. The embryos were washed with 100% HYB, 50% HYB/2× SSC supplemented with 0.1% Tween 20 (2× SSCTw), and 2× SSCTw at 65 °C for 15 min each, followed by two washes with 0.2× SSCTw at 65 °C for 30 min. Then, the embryos were treated with 1× MABDT (1× maric acid buffer (MAB), 1% DMSO, and 0.1% Tween 20) at RT for 15 min and a blocking solution (1% Blocking Reagent (Merck catalog no. 11096176001), 1× MAB, 10% FBS, 0.1% Tween 20, 1% DMSO) for 1 h at room temperature. Then, the embryos were incubated with anti-digoxigenin-AP Fab fragments (Roche, Basel, Switzerland, catalog no. 11093274910) diluted 1/5000 in blocking solution at RT for 3 h. The embryos were washed with 1× PBSTw eight times at RT for 15 min and held at 4 °C overnight. The embryos were washed with NTMT (100 mM NaCl, 500 mM MgCl_2_, 100 mM Tris-HCl, pH 9.5, and 0.1% Tween 20) twice at RT for 5 min and stained with coloration solution (3.5 µL of BCIP solution (Merck, catalog no. 11383221001) and 2.3 µL of NBT (Roche catalog no. 11483213001) per 1 mL of NTMT) for approximately 2 h to overnight at RT or 4 °C in the dark. After color development was complete, the embryos were washed with 1× PBSTw three times for 5 min at RT and stored in 4% PFA/1× PBS at 4 °C. Images were captured using a stereomicroscope (Leica MZ16) and Axio Cam MRc (Zeiss). The measurement of the angle of the *chrd* expression domain was performed using Fiji software [51].

### 4.12. Alizarin Red and Alcian Blue Staining

The fertilized eggs were dechorionated at 1 dpf and treated with DMSO, NPL1010 (2.5 or 5 μM in E3 medium), NPL3008 (2.5 or 5 μM in E3 medium), or dorsomorphin (5 μM in E3 medium) until 5 dpf. Staining was performed as described (Walker and Kimmel, 2007). Briefly, 5 dpf embryos were collected in a 1.5 mL tube and fixed with 4% PFA for 2 h. Then, the embryos were dehydrated with 50% ethanol at room temperature for 10 min and stained with staining solution (0.02% Alcian blue, 0.05% Alizarin red, 200 mM MgCl_2_, and 70% EtOH) at room temperature overnight. After staining, embryos were treated with 1.5% H_2_O_2_ and 1% KOH at room temperature for 20 min. Then, the embryos were cleared by sequential treatment with 0.375% KOH, 25% glycerol; 0.375% KOH, 50% glycerol; and 0.125% KOH, 75% glycerol at room temperature for 1–3 days and stored in 100% glycerol at 4 °C. Images were captured using a stereomicroscope (Leica MZ16) and Axio Cam MRc (Zeiss).

### 4.13. mRNA Synthesis and Phenotypic Rescue Experiment

The *bmp2b*, *bmp1a*, *tll1*, and *chrd* mRNAs were generated from pCS2 + kzbmp2, pCS2 + bmp1a [11], pCS2 + tll1 [52], and pCS2 + chrd-myc [48] by restriction digestion with NotI and AmpliCap SP6 High Yield Message Maker Kit (CellScript, Madison, WI, USA). The synthesized capped mRNA was purified using a SigmaSpin™ PostReaction Clean-Up Column (Merck) and stored at −80 °C. The generated mRNA was injected into zebrafish embryos at the one-cell stage using an IM300 microinjector (NARISHIGE, Tokyo, Japan). In the rescue experiment, the mRNA injected embryos were exposed to the compounds at 3–24 hpf as described above, and the phenotype was observed.

### 4.14. Molecular Docking

First, the structure of zebrafish BMP1 (BMP1_DANRE) for the docking study was modeled using the SWISS-MODEL server with the human BMP1 structure as the template (PDB: 6BSL). Then, hydrogen atoms were added and subjected to further docking experiments with AutoDock Vina v1.1.2 according to the instructions [53]. The grid box size was defined around a known BMP1 inhibitor (hydroximate Compound 22) binding site with dimensions of 40 × 40 × 40 Å grid points. Ligand conformations of NPL1010 and NPL3008 were obtained from the virtual screening database LIGANDBOX (version 1306) [54]. Molecules were docked using AutoDock Vina with exhaustiveness grade 8 and a maximum of 100 poses searched per molecule. The lowest energy conformations for NPL1010 and the second-lowest energy conformation for NPL3008, which exhibited good agreement with NPL1010 binding, were selected. The results were visualized using PyMOL (opensource version, Schroedinger).

### 4.15. Interaction Simulation

AlphaFold simulation was performed using AlphaFold Colab notebook on Google Colaboratory [42,43]. The results were visualized using PyMOL.

### 4.16. Statistical Analysis and Graphing

Statistical analyses were performed using Prism 8 (GraphPad Software). Information about error bars, sample size, and statistical test methods is described in the corresponding figures and figure legends.

## Figures and Tables

**Figure 1 ijms-24-04313-f001:**
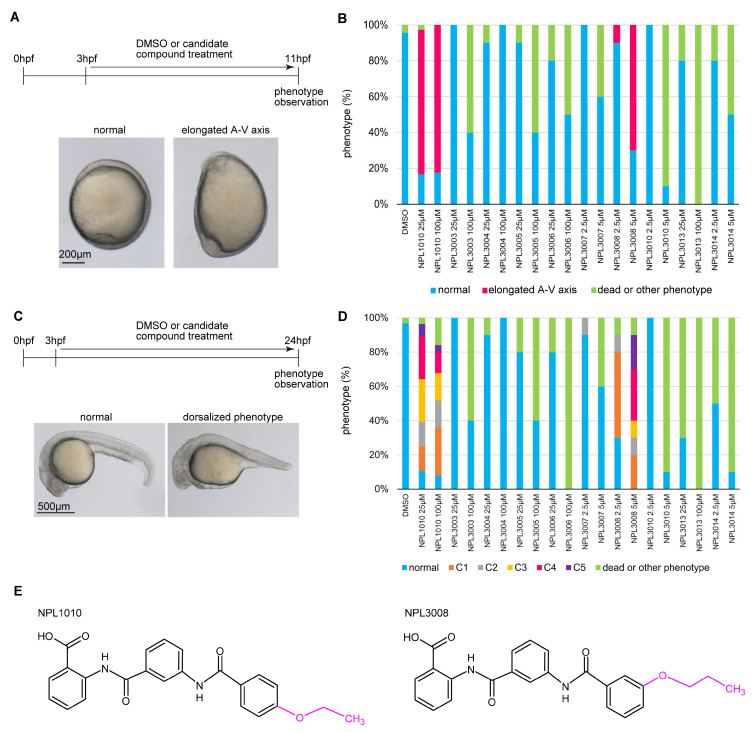
Screening chemical compounds using zebrafish embryos. (**A**) Schematic diagram of the phenotype screen performed at 11 hpf (upper panel), with the representative normal and elongated A-V axis phenotypes shown in the photographs (bottom panels). Scale bar, 200 μm. (**B**) The proportion of phenotypes observed in zebrafish treated with compounds at 11 hpf (*n* = 10–45). (**C**) Schematic diagram of the phenotype screen at 24 hpf (upper panel) with the representative normal and dorsalized phenotypes shown in the photographs (upper panel). Scale bar, 500 μm. (**D**) The proportion of zebrafish presenting the dorsalized C1−C5 phenotypes after treatment with compounds at 24 hpf (*n* = 10–45). (**E**) The molecular structures of NPL1010 and NPL3008. These compounds share a common structure (black line and letters), but the ether group length and position differ (magenta line and letters).

**Figure 2 ijms-24-04313-f002:**
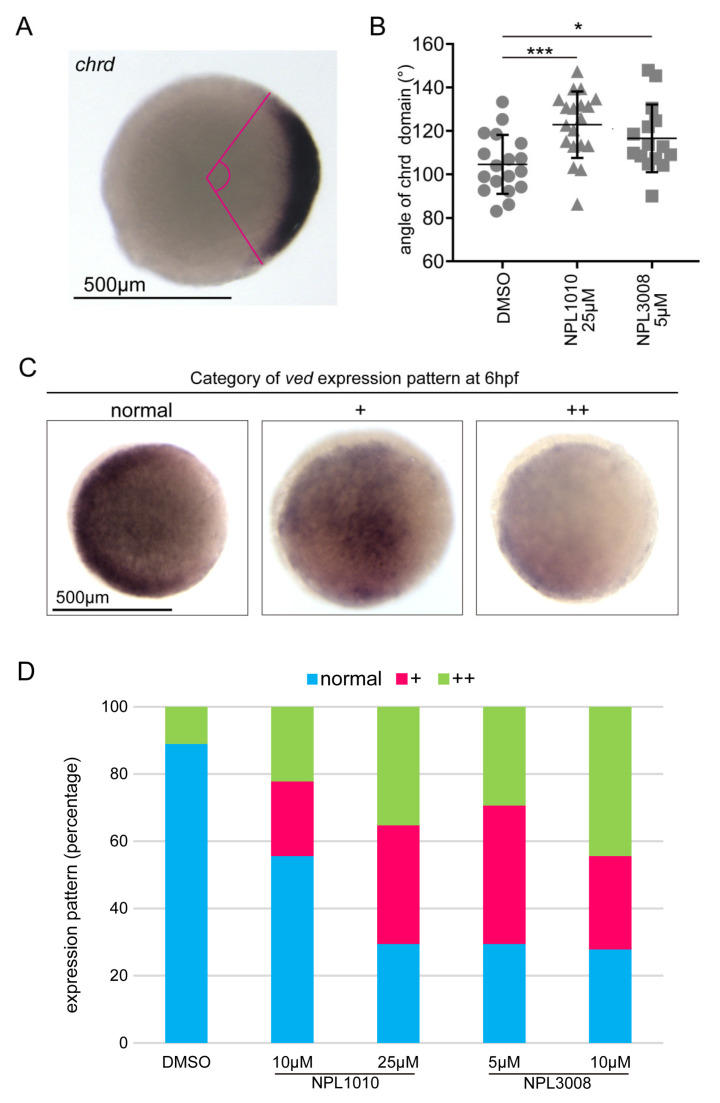
NPL1010 and NPL3008 affect the expression of genes involved in D–V patterning. (**A**) Method used to measure the angle of the *chrd* expression domain with representative *chrd* expression shown in the photographs. The central angle (shown by the magenta line) of the *chrd* expression domain was measured. Animal pole view, dorsal to the right. Scale bar, 500 μm. (**B**) The statistical analysis of angles of the *chrd* expression domain in NPL1010- or NPL3008-treated embryos. Data are presented as the means ± s.d., with individual data shown. *: *p* < 0.05, ***: *p* < 0.001 (Dunnett’s test). (**C**) Classification of the *ved* expression pattern, normal, slight reduction/mild phenotype (+), and strong reduction/severe phenotype (++), at 6 hpf with representative expression patterns shown in the photographs. Animal pole view, ventral to the left. Scale bar, 500 μm. (**D**) Proportion of NPL1010- or NPL3008-treated embryos displaying *ved* expression patterns (*n* = 17–18).

**Figure 3 ijms-24-04313-f003:**
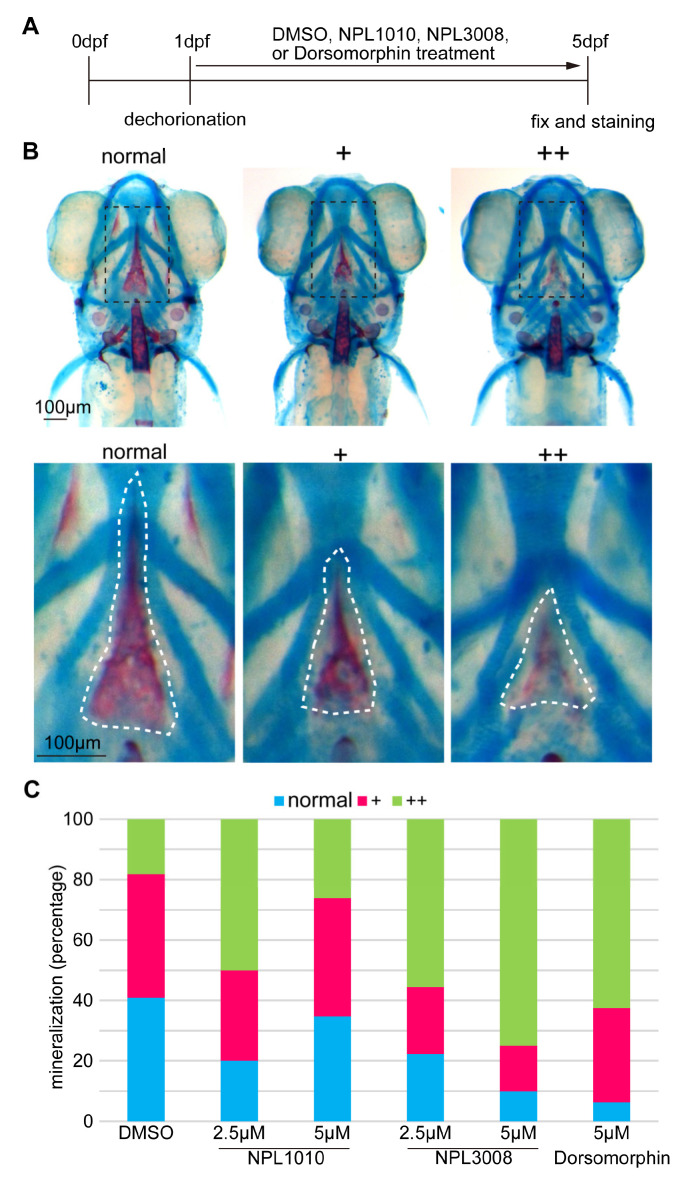
NPL1010 and NPL3008 inhibit bone formation in vivo. (**A**) Schematic diagram of the analysis of the effects of NPL1010 and NPL3008 on bone formation in zebrafish embryos. (**B**) Images of Alizarin red and Alcian blue staining and classification of bone mineralization: normal, slight reduction/mild phenotype (+), and strong reduction/severe phenotype (++). Dorsal views of the head, anterior to the top (upper panels). Red staining shows mineralized bone, and blue staining shows cartilage. The black dotted boxed region indicates the parasphenoid bone region. Scale bar, 100 μm. Enlarged views of the boxed area in the upper panels are shown in the bottom panels. The white dotted lines indicate mineralized parasphenoid bone. Scale bar, 100 μm. (**C**) Proportion of zebrafish displaying phenotypes after treatment with NPL1010, NPL3008, and dorsomorphin (each category corresponds to (**B**)) (*n* = 9–23).

**Figure 4 ijms-24-04313-f004:**
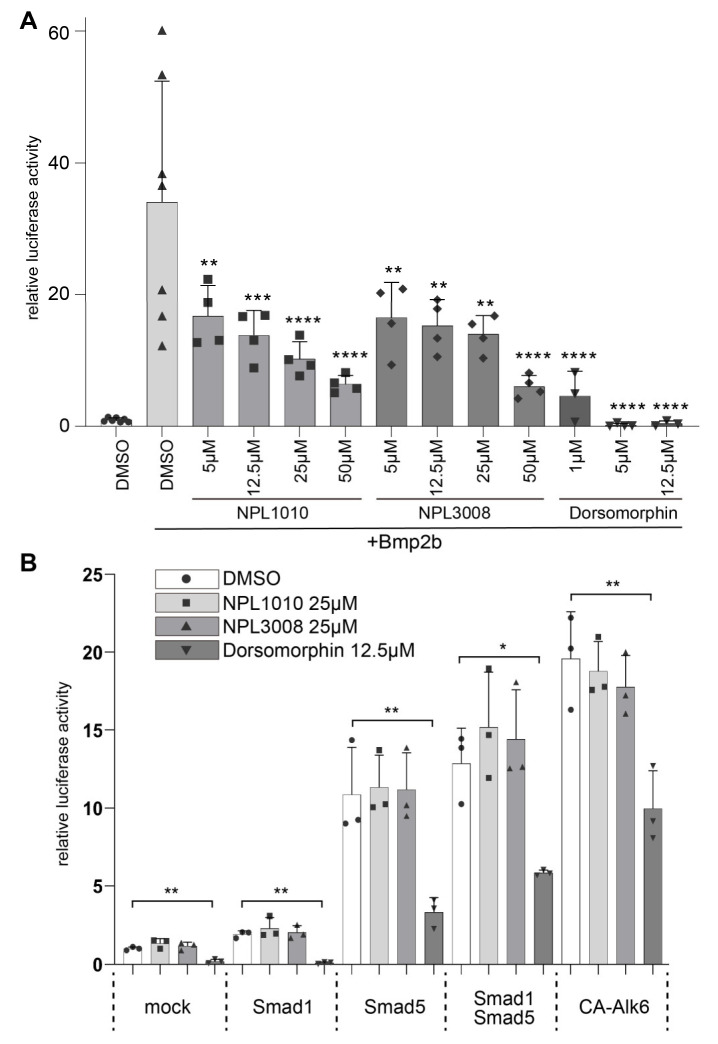
NPL1010 and NPL3008 decrease activity of the BMP signaling reporter. (**A**) Relative luciferase activity of the BMP signaling reporter. BRE reporter and internal control plasmids were transfected into C2C12 cells. The control or *bmp2b* expression vector was cotransfected, and cells were treated with the indicated concentrations of NPL1010, NPL 3008, or dorsomorphin. The cell lysate was collected, and luciferase activity was measured. Luciferase activities relative to the activity in control-vector-transfected and DMSO-treated cells are shown. **: *p* < 0.01, ***: *p* < 0.001, and ****: *p* < 0.0001 (Dunnett’s test). The error bars indicate +s.d., with individual data shown (*n* = 3–7). (**B**) Relative luciferase activity of the BMP signaling reporter. Overexpression of CA-Alk6, Smad1, and Smad5 increased BRE reporter-dependent luciferase activity in C2C12 cells. In cells overexpressing any combination of these genes, NPL1010 and NPL3008 did not induce a significant decrease in luciferase activity compared to the DMSO control, although dorsomorphin significantly reduced luciferase activity. *: *p* < 0.05, **: *p* < 0.01 (Dunnett’s test). Luciferase activities relative to the activity in control-vector-transfected and DMSO-treated cells are shown. The error bar indicates +s.d. with individual data shown (*n* = 3).

**Figure 5 ijms-24-04313-f005:**
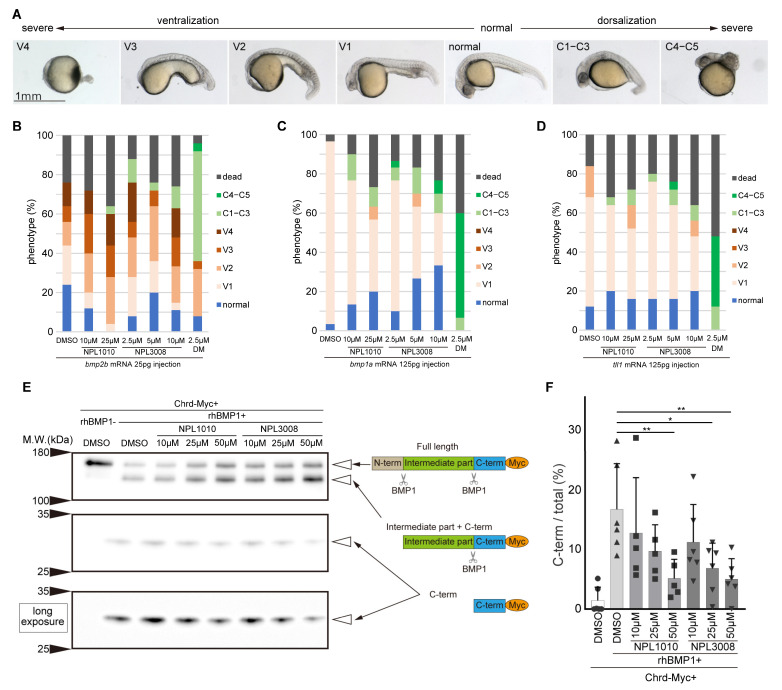
NPL1010 and NPL3008 inhibit BMP1-dependent Chrd cleavage. (**A**) Classification of ventralization and dorsalization phenotypes with representative phenotypes shown in the photographs. The phenotypes were classified into ventralized (V1−V4), normal, and dorsalized (C1−C5). Lateral view, anterior to the left. Scale bar, 1 mm. (**B**–**D**), Proportion of *bmp2b* (**B**), *bmp1a* (**C**), or *tll1* (**D**) mRNA-injected embryos displaying the phenotype. DM: dorsomorphin. NPL1010 and NPL3008 partially rescue the *bmp1a*- or *tll1*-induced ventralization phenotype (*n* = 25–30). (**E**) Chrd cleavage assay. Chrd-Myc was incubated with rhBMP1. The effects of NPL1010 and NPL 3008 on BMP1-dependent Chrd cleavage were analyzed using Western blotting. The schematic illustration shows the cleavage patterns of Chrd. (**F**) Statistical analysis of the amount of cleaved Chrd detected in the Chrd cleavage assay. NPL1010 and NPL3008 dose-dependently inhibited BMP1-dependent Chrd cleavage. *: *p* < 0.05 and **: *p* < 0.01 (Dunnett’s test). Error bars indicate +s.d. with individual data shown (*n* = 5–6).

**Figure 6 ijms-24-04313-f006:**
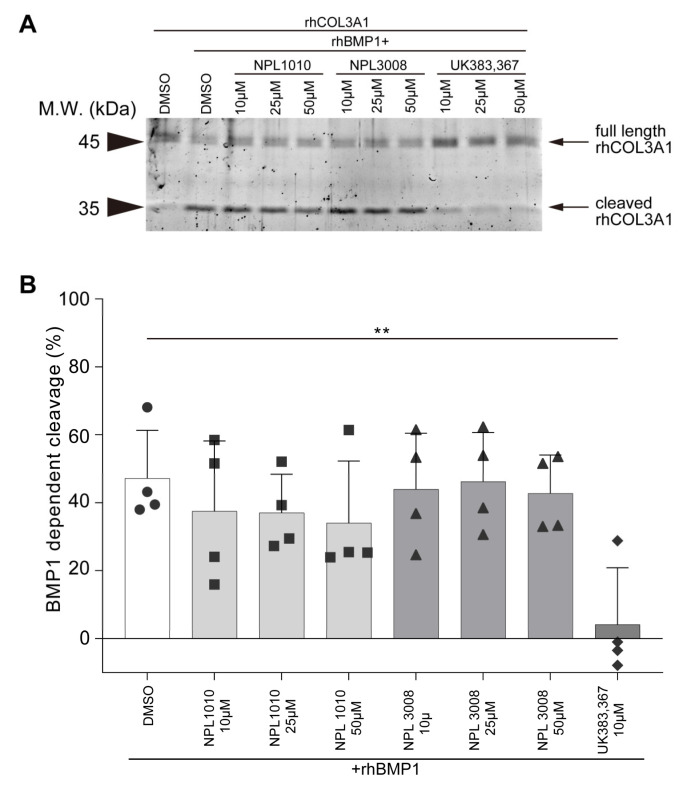
NPL1010 and NPL3008 rarely inhibit BMP1-dependent COL3A1 cleavage. (**A**) COL3A1 cleavage assay. rhCOL3A1 was incubated with rhBMP1. The effects of NPL1010 and NPL3008 on BMP1-dependent rhCOL3A1 cleavage were analyzed using SDS–PAGE and Oriole staining. Arrows indicate full-length rhCOL3A1 and cleaved rhCOL3A1. (**B**) Statistical analysis of BMP1-dependent cleavage of COL3A1. The levels of noncleaved and cleaved COL3A1 were measured by performing densitometry of the Oriole-stained gel, and the percentage of cleaved COL3A1 was calculated based on the fragments detected at each molecular weight and background. **: *p* < 0.01 (Dunnett’s test). The error bar indicates +s.d. with individual data shown (*n* = 4).

**Figure 7 ijms-24-04313-f007:**
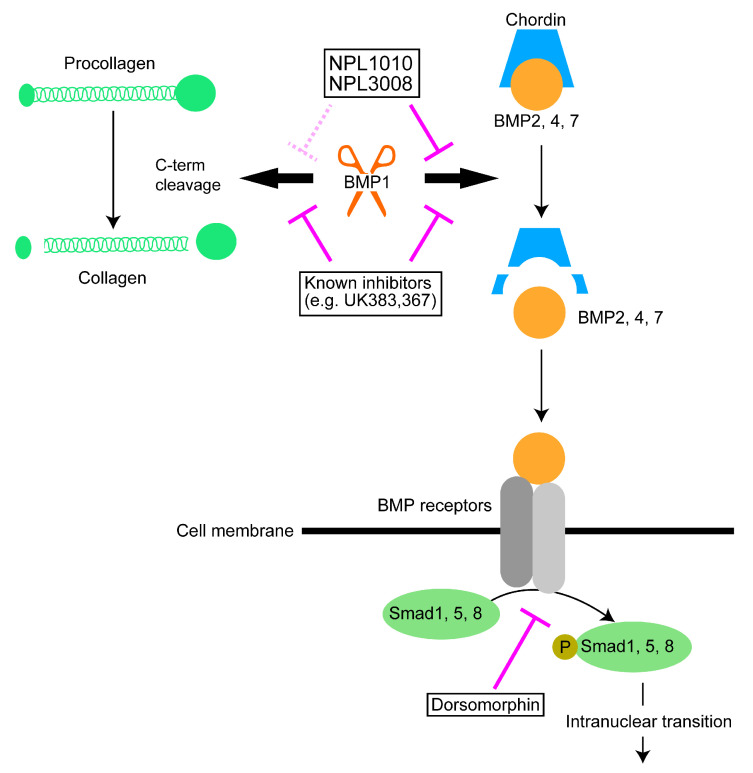
Hypothetical model of NPL1010 and NPL3008 function.

## Data Availability

The data used to support the findings of this study are available from the corresponding author upon request.

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
