# Peer review of "Small-Molecule-Mediated Suppression of BMP Signaling by Selective Inhibition of BMP1-Dependent Chordin Cleavage"

_ijms, 2023, doi:10.3390/ijms24054313_

Round 1

Reviewer 1 Report

In this study, the authors identified NPL1010 and NPL3008 as potential regulators for Wnt and BMP signaling. They used zebrafish embryos to confirm that these chemicals caused ventrolization via inhibiting Bmp1 function, as a result repressed the cleavage of Chrd to favor Chrd inhibiting BMP signaling. The study is carefully designed and well-written.

Comments:

Fig. 1. The authors created a term, "football shape", to describe the elongated animal-vegetal axis phenotype. This phenotype seemly caused by convergent/extension defects during gastrulation. I would suggest to use convergent/extension defects or elongated animal-vegetal axis to describe this phenotype instead of "football shape".

Fig. S5. myoD and krox20 are markers for segmentation and not related to D-V patterning. Perhaps the authors can provide more explanation for the reason of analyzing these markers.

Fig. 2. The quality of in situ hybridization, especially ved, needs to be improved.

Fig. 2. Measuring the angle of expression from animal pole rely on a subjective observation of each examiner and is also highly depend on the orientation of the embryo spheres. Therefore, a better quantification method, such as qPCR, is required to measure chrd and ved expression.

Fig. 2 and Fig. 3. The authors used normal (+++), slight reduction (++), and strong reduction (+) to describe different severities of the phenotypes. I would suggest simply use normal (without symbol) for unaltered phenotype, and number of symbols for the severities of phenotype, e.g. slight reduction/mild phenotype (+), and strong reduction/severe phenotype (++).

Fig. 7 needs a figure legend.

There are many grammar and typographic errors. I would recommend a thoroughly English editing.

Author Response

Thank you for your careful review of our manuscript.
We provide a point-by-point response to the reviewer’s comment as an attached file.

Point-by-point response to the reviewer’s comments

Comment1
Fig. 1. The authors created a term, "football shape", to describe the elongated animal-vegetal axis phenotype. This phenotype seemly caused by convergent/extension defects during gastrulation. I would suggest to use convergent/extension defects or elongated animal-vegetal axis to describe this phenotype instead of "football shape".

Response 1

As the reviewer suggested, we changed ” football shape” to “elongated animal-vegetal (A-V) axis phenotype”.

Comment 2

Fig. S5. myoD and krox20 are markers for segmentation and not related to D-V patterning. Perhaps the authors can provide more explanation for the reason of analyzing these markers.

Response 2

As the reviewer pointed out, our previous explanation was confusing. We used these markers to confirm the expansion of dorsal structure as previously reported (Little and Mullins 2004; Tucker, Mintzer, and Mullins 2008). We added the explanation and citation in line 126-127, “To confirm the expansion of dorsal structure by changes in gene expression patterns, as previously reported [21,22],”.

Comment 3

Fig. 2. The quality of in situ hybridization, especially ved, needs to be improved.

Response 3

As the reviewer suggested, we changed the figures of ved expression in Figure 2C.

Comment 4

Fig. 2. Measuring the angle of expression from animal pole rely on a subjective observation of each examiner and is also highly depend on the orientation of the embryo spheres. Therefore, a better quantification method, such as qPCR, is required to measure chrd and ved expression.

Response 4

In this experiment, we focus on the change of marker gene expression domain. Although the reviewer suggested qPCR analysis, qPCR cannot reveal the marker gene expression patterning. Therefore, we performed the in situ hybridization and angle measurement of dorsal expression domain of chrd. Angle measurement was previously used to assess expansion of dorsal expression domain (Feng et al. 2014). We aligned the orientation of samples properly and took pictures. Considering the orientation and spheres variation of each sample, we performed statistical analysis, and confirmed that the chrd expression region was significantly expanded in the NPLs-treated group. In addition, it was reported that dorsliztion reduces the expression of ved so that it can be distinguished by in situ hybridization(Shimizu et al. 2002; Ye et al. 2019). However, our previous manuscript was confusing; therefore, we added the explanation and citations in line 139-141, ” In dorsalized embryos, dorsal expression domain of chrd is expanded[21,25,26]. Therefore, we measured the central angle of the chrd expression domain as previously reported [25].”, and line 148-149 “Previous reports indicated that ved and szl expression domain was reduced in dorsalized embryos [29,30].”.

Comment 5

Fig. 2 and Fig. 3. The authors used normal (+++), slight reduction (++), and strong reduction (+) to describe different severities of the phenotypes. I would suggest simply use normal (without symbol) for unaltered phenotype, and number of symbols for the severities of phenotype, e.g. slight reduction/mild phenotype (+), and strong reduction/severe phenotype (++).

Response 5

As the reviewer suggested, we changed the description of the severities of the phenotypes, and corrected figures 2 and 3.

Comment 6

Fig. 7 needs a figure legend.

Response 6

As the reviewer suggested, We added the legend of Fig7.

Comment 7

There are many grammar and typographic errors. I would recommend a thoroughly English editing.

Response 7

As the reviewer suggested, we conducted English editing through an English editing service.

Reviewer 2 Report

The authors designed an experiment to investigate the inhibition of BMP1 signaling by NPL1010 and 3008. 

some questions need to be addressed in respective part of the manuscript which might be interesting for readers.

(1) Could the authors hypothesize about the selectivity of inhibition (based on the results)? Is the inhibition competitive? reversible? Are there any other results?

(2) is the inhibition of BMP1 signalling sufficient to achieve the therapeutic goals (e.g. by the treatment of depression phenotype, the aging dependent decline in cognitive functions?

(3) The authors used the zebrafish model. Are the results transferable to humans in terms of effect, efficiency, signaling and pharmacology?

Author Response

Thank you for your careful review of our manuscript.
We provide a point-by-point response to the reviewer’s comment as an attached file.

Reviewer 2

some questions need to be addressed in respective part of the manuscript which might be interesting for readers.

Comment2-1

(1) Could the authors hypothesize about the selectivity of inhibition (based on the results)? Is the inhibition competitive? reversible? Are there any other results?

Response 2-1

Our simulation results suggest that NPL1010 and NPL3008 might interact with the putative Chrd interacting site of the BMP1 catalytic domain. Therefore, NPL1010 and NPL3008 might function as reversible and competitive inhibitors. To address this possibility, further studies are needed. We added this discussion in line365-369, “Our simulation results suggest that NPL1010 and NPL3008 might interact with the putative Chrd interacting site of the BMP1 catalytic domain. Therefore, NPL1010 and NPL3008 might function as reversible and competitive inhibitors. To address this possibility, further studies are needed.”.

Comment2-2

(2) is the inhibition of BMP1 signalling sufficient to achieve the therapeutic goals (e.g. by the treatment of depression phenotype, the aging dependent decline in cognitive functions?

Response 2-2

Selective inhibition of BMP1-mediated Chrd cleavage might be a useful therapeutic strategy. The Chrd level is decreased in the hippocampus of a mouse model of chronic social defeat stress (CSDS), and compensation of Chrd rescues the CSDS-dependent depression phenotype (Wang et al., 2020). Since BMP1 is also expressed in the hippocampus (http://www.proteinatlas.org) (Uhlen et al., 2015), inhibition of BMP1-mediated Chrd cleavage might suppress CSDS-dependent depression. Moreover, an increase in BMP4 expression with age in the mouse hippocampus is associated with decreased neurogenesis and diminished hippocampus-dependent cognitive functions. Inhibition of BMP signaling in the hippocampus of aged mice restores cognitive function to levels comparable to young mice (Meyers et al., 2016). Thus, NPL1010 and NPL3008 might be able to treat brain-related abnormalities such as depression and dementia by decreasing BMP signaling through the inhibition of BMP1-dependent Chrd cleavage.

We added this possibility in line 385-390, ”We demonstrated that NPL1010 and NPL3008 can prevent human BMP1-dependent Chrd cleavage. Although the efficiency and pharmacokinetics of NPL1010 and NPL3008 in human body must be investigated, NPL1010 and NPL3008 may be applied in the treatment of abnormal BMP signaling causing human diseases, such as the depression phenotype and the aging-dependent decline in cognitive function, by upregulating the functional Chrd protein level [4–6]. ”.

Comment2-3

(3) The authors used the zebrafish model. Are the results transferable to humans in terms of effect, efficiency, signaling and pharmacology?

Response 2-3

BMP signaling is conserved among vertebrate and our data showed that NPL1010 and NPL3008 can prevent human BMP1-dependent Chrd cleavage. However, efficiency or pharmacokinetics must be addressed in future studies. We added this discussion in line 385-390 as described in Response 2-2.

Round 2

Reviewer 1 Report

the authors have fully addressed each questions and I suggest this article is now ready for publication.